# Generation of High-Frequency Ultrasound in a Liquid upon Excitation by Laser Radiation through a Light Guide with a Converter of Transparent Spheres

**Vladimir I. Bredikhin** *[ID] and **Viacheslav V. Kazakov** *

Department of Nonlinear Dynamics and Optics, Institute of Applied Physics of the Russian Academy of Sciences (IAP RAS), 46 Ulyanov Street, 603950 Nizhny Novgorod, Russia
* Correspondence: bredikh@ipfran.ru (V.I.B.); kazak@ipfran.ru (V.V.K.)

**Abstract:** One of the important tasks in optoacoustics today is the development of methods and tools for generating high-frequency ultrasound (above 1 MHz) in liquids and other media. To expand the frequency range of ultrasound, it was proposed to use coatings consisting of focusing spheres on a fiber tip. The methodology of calculating the ultrasound spectra depending on the sphere size, index of refraction, and parameters of laser radiation was developed. Two cases of small and large spheres in strongly and weakly absorbing media were simulated. The experimental results were analyzed in the approximations allowing a fairly accurate estimation of the spectrum and indicatrix of the generated ultrasound upon laser excitation through a converter based on a coating of transparent spheres. A good agreement between the model and experimental result was obtained.

**Keywords:** 2D colloidal coating; optical fiber; laser radiation converter; polystyrene (1 μm) and glass (200 μm) spheres; optoacoustics; laser ultrasonics; generation of high-frequency ultrasound

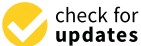



## 1. Introduction

The development of methods and tools for generating high-frequency (above 1 MHz) ultrasound (US) in liquids and other media is highly important for applications in medicine, nanotechnology, information processing, and other areas [1]. The intensity of ultrasound must be high enough to ensure effective localized impact on materials, including biological ones. Modern laser devices can provide sufficiently powerful optical radiation with virtually any optical wavelength and time spectrum (modulation frequency). The only necessary condition is to use the "replica" of the time spectrum of optical radiation for the acoustic range.

Laser excitation of ultrasound in various media has been well-studied and widely used. Suffice it to mention the informative articles and reviews [1–5]. It is used, for example, for coherent optical signal processing in multichannel acoustic information systems, for non-contact remote optical methods of research, for recording acoustic fields and vibrations, for acoustic receivers (fiber-optic sound receivers), for coherent optical computer systems, for methods and devices of non-destructive control, and for improving the physicochemical properties of materials [1–5].

Among this variety of applications of optoacoustic (OA) methods, the techniques using fiber-optic laser devices rank high. They make it possible to introduce exciting laser radiation in the simplest, practically non-destructive way to various parts of devices, tissues, and organs of the body. Recently, an interest in this method of ultrasound excitation, especially in biological tissues, has been renewed, since it was found that high-frequency megahertz ultrasound can be used to improve drug delivery and suppress the activity of microorganisms [6,7]. However, no means of generating a powerful ultrasound in the range above 1 MHz have been proposed yet.

Naturally, for effective suppression of microorganisms, the generated ultrasound, in addition to high frequency (several MHz), must be sufficiently intense. For this, it is logical to focus the radiation emerging from the fiber, for example, using transparent microlenses or transparent microspheres deposited on the fiber end [8,9]. No detailed enough research aimed at testing and exploiting such ideas has been reported in the literature yet. In our earlier works [8–11], we carried out experimental studies in two relatively limiting cases: large spheres with a diameter of 200 μm in a highly absorbing medium (ink solution in water at a wavelength of 0.532 μm, light absorption coefficient $\alpha \approx 10^{-2}$ μm$^{-1}$), and small spheres with a diameter of 0.96 μm in a weakly absorbing medium (pure water at a wavelength of 1.064 μm, $\alpha \approx 10^{-5}$ μm$^{-1}$). The considered media were chosen because they are the simplest phantoms of biological media.

A detailed theoretical analysis of the experimental results, allowing generalization of the data for other possible parameters, has not been performed. Below, we carry out such an analysis in the approximations that make it possible to fairly accurately estimate the efficiency and nature of the generated ultrasound on laser excitation through a converter consisting of a system of transparent spheres.

A methodology for calculating the main characteristics of the generated ultrasound (the frequency spectrum and the angular diagram) is presented (Part 2). Part 3 (Results) contains the results of calculating US parameters in two opposite cases: large spheres with a diameter of 200 μm in a highly absorbing medium (ink solution in water at a wavelength of 0.532 μm, light absorption coefficient $\alpha \approx 10^{-2}$ μm$^{-1}$) and small spheres with a diameter of $\approx 1$ μm in a weakly absorbing medium (pure water at a wavelength of 1.064 μm, $\alpha \approx 10^{-5}$ μm$^{-1}$), as well as comparison of the results of calculations with the corresponding experimental results from the papers [8,9]. Part 4 (Discussion) is devoted to the analysis of the comparison of theoretical and experimental results both from the point of view of the applicability of the theoretical consideration and from the viewpoint of the possibilities and prospects for the development of ultrasound generation technology using fiber systems coated by transparent spheres. Part 5 (Conclusions) summarizes the basic results of the presented work, highlighting unresolved scientific and technical problems in the development and application of new tools in surgery and other fields of science and technology.

## 2. Analysis of Thermo-Optical Sound Excitation Process

The generation of ultrasound excited by a laser pulse has been well studied in the literature [2–5] and is described by two equations—the wave equation for the propagation of sound pressure $p$ (1) and the heat Equation (2):

$$\Delta p - c^{-2}\frac{\partial^2 p}{\partial t^2} - 2\Gamma\frac{d(\Delta p)}{dt} = -\xi\frac{\partial^2 T}{\partial t^2}, \tag{1}$$

$$\frac{\partial T}{\partial t} = -\chi\Delta T + \frac{\alpha}{\rho c_p}I(r,t), \tag{2}$$

where $c$ is the speed of sound in the medium, $\Gamma = \gamma/k\omega$, $\gamma$ is the sound absorption coefficient, $\xi$ is the coefficient of volumetric thermal expansion, $k$ and $\omega$ are the wave vector and sound frequency, $T$ is the temperature, $\chi$ is the thermal diffusivity, $c_p$ is the specific heat capacity, $I(r,t)$ is the laser radiation intensity, $r$ is the coordinate, and $t$ is the time.

At laser excitation of ultrasound, the characteristic times and the corresponding frequencies $\nu = \omega/2\pi$ of the processes in Equations (2)–(5) are the following: $\nu_1 = \tau_1^{-1}$, where $\tau_1$ is the travel time of sound through the irradiated heated area $\tau_1 = a/c$ and $\nu_2 = \tau_2^{-1}$, where $\tau_2$ is the characteristic cooling time [12] of the heated region $\tau_2 = a^2/\chi$ (for Equation (2)), where $a$ is the characteristic size of the laser beam.

The parameter $2\Gamma$ determined by the attenuation of ultrasound is, as a rule, much less than the frequencies $\nu_1$ and $\nu_2$ [1–5].

Since system (1-2) is linear, by expanding $I(r, t)$ into a Fourier series $I(r, \nu_m)$, where $\nu_m$ is the light modulation frequency (not to be confused with the carrier frequency), it is possible to consider the response–pressure $p(r, \nu)$ at the same frequency $\nu = \nu_m$. Obviously, the nature of the excitation process $p(r, \nu)$ depends on the ratio of the frequencies $\nu$, $\nu_1$ and $\nu_2$, and hence on the laser beam size $a$.

### 2.1. Thermo-Elastic (TE) Excitation of Ultrasound

In most experimental situations, $\nu \sim \nu_1 >> \nu_2$; in this case $\partial T / \partial t \approx \alpha / (\rho c_v)\, I(r, t)$ and system (1), (2) reduces to Equation (3) (thermo-elastic mechanism):

$$\Delta p - \mathrm{c}^{-2} \frac{\partial^2 p}{\partial t^2} = - \frac{\xi \cdot \alpha}{c_p} \frac{\partial I(r,t)}{\partial t} \tag{3}$$

which is studied most often [2–5].

The sound pressure spectrum $p(\nu, r, \theta)$ upon excitation by a parallel Gaussian laser beam having radius $a$ with intensity spectrum $J_0(\nu)$ according to [2–4] has the form:

$$p(r, \nu, \theta) = - \frac{2\pi\nu\, AJ_0(\nu)a^2}{2c_p} \times \frac{e^{(jkr)}}{r} \times \frac{\alpha\left(2\pi\nu/c\right)\cos\theta}{\alpha^2 + \left(2\pi\nu/c\right)^2 \cos\theta^2} e^{-\left(\frac{\pi^2 a^2 \nu^2}{c^2}\sin\theta^2\right)}, \tag{4}$$

where $A$ is the coefficient of light transmission through the liquid boundary, $\theta$ is the angle between the direction of incidence of the laser beam and the direction from the observation point to the entry point of the laser beam, and $r$ is the distance from the observation point to the entry point.

It can be seen that a rather complex angular-frequency structure of the generated ultrasound and its dependence on the light absorption coefficient of the medium physically indicate the result of the interference of the ultrasonic fields generated by elementary volumes of the medium heated by optical radiation. We kept this circumstance in mind when considering and interpreting the experimental results obtained when ultrasound is excited by more complex light fields.

It follows from solution (4) that the spectrum of the OA response $p(\nu)$, on the whole, coincides with the excitation spectrum $J(\nu)$; no new frequencies arise in this case. Ultrasonic signals with frequencies $\nu$ below $\nu_1$ can be actively generated, i.e., in channels with radius $r < c/\nu$, of course, if such frequencies are present in the exciting laser signal. For $\nu_1 < \nu$, the excitation efficiency drops sharply (at a large diameter of the exciting beam). With a decrease in the beam radius, the frequency generation range extends to higher frequencies (by analogy with a string–the thinner the channel, the higher the sound frequency), and the angular diagram of ultrasound generation extends to larger angles.

The above consideration leads to the following conclusions:

Narrowing of the exciting beam (focusing) leads to the expansion of the ultrasonic generation range to the high-frequency region and to the restructuring of the directivity pattern with signal amplification at $\theta > 0$.

Narrowing of the exciting beam allows generating the US in the high-frequency region $\nu \leq \nu_1 \sim r^{-1}$.

It might seem that splitting of the exciting beam of radius $a_0$ into separate $N^2$ beams with radii $a \sim > a_0/N$, followed by separate focusing, allows expanding the range of ultrasonic generation with the upper-frequency $\nu \sim c(a_0/N)^{-1}$. However, this is not quite true. The point is that, due to the interference of the ultrasonic waves generated at different distances from the beam center, there appears a factor of the type $\frac{\sin(2\pi ka/N)^2}{(2\pi ka/N)^2}$ which reduces the US intensity at $k \cdot a > 1$.

### 2.2. Thermal Relaxation Mechanism of US Excitation

In the case ($\nu \leq \nu_2 << \nu_1$), when the signal frequency is much less than the characteristic frequency $\nu_1$, the pressure is determined by heating of the irradiated region during the

pulse time $\tau$ and cooling during the characteristic time $\sim \nu_2^{-1}$ (thermal relaxation (TR) mechanism of US excitation). The characteristic temperature relaxation frequency is $\nu_2 \sim \chi/a^2$ [12], $\chi$ is the thermal diffusivity, and $a$ is the characteristic size of the excited region (separate beam). That is why the condition of the TR mechanism operation is $a \leq \chi/c$ [2–4]. For the characteristic values $\chi \sim 0.14 \times 10^{-6}$ m$^2$/s and $c \sim 1.5 \times 10^3$ m/s (water), we obtained $a \leq 10^{-3}$ µm, which is three orders of magnitude smaller than the minimal diameter of the laser beam in the caustic of the spherical lens (see, e.g., Section 3). With a beam diameter of 1 µm, the maximum frequency of the TR influence is at $\nu_2 \sim \chi/a^2 \leq 100$ kHz, which is beyond the considered frequency range. Therefore, in what will follow, the TR mechanism is not considered.

The above analysis was made in a linear approximation. With sharp focusing of the beam in the region of high intensities, nonlinear and polyphase processes can occur, for example, the generation of microbubbles with subsequent cavitation [7]. These processes are very important, but their detailed consideration is beyond the scope of this consideration; for their search and analysis, nonlinear effects should be addressed.

In the next sections, based on the qualitative analysis made above, we present results of the quantitative consideration of US generation using coatings from transparent spheres and compare them with the experimental data [7–9] on the generation of high-frequency US with excitation through a layer of focusing microspheres.

### 2.3. The Main Factors Determining the Properties of Laser Generation of Ultrasound When Laser Radiation Is Injected through a Layer of Focusing Spheres

When considering the generation of ultrasound in the case of laser radiation injection through a layer of spheres, the following points should be taken into account:

Laser beam structure in the medium;

Beam structure from a single sphere. This problem was analyzed in detail in [11] for spheres with a diameter larger than the laser wavelength;

Beam structure from a system of spheres. For a system of small spheres (0.96 µm), such an analysis was carried out in [9]. It was shown that, in a weakly absorbing medium, a laser beam breaks up into a three-dimensional interference periodic system of prolate spheroids, the minimal size of which (and, consequently, the maximal light intensity) is determined by beam focusing by a single sphere.

In the case of focusing on large spheres in a strongly absorbing medium, the light structure mainly consists of rays formed by individual spheres [8]. In this case, the following points should be considered:

Ultrasound generated by a beam after a single sphere;

Interference of ultrasound generated by a periodic system of laser beams.

In further analysis, we will mainly adhere to the above scheme.

### 2.4. Ultrasound Generation upon Laser Excitation through a Layer of Focusing Spheres

2.4.1. Large Spheres, Environment with High Light-Absorption

Consider the generation in a strongly absorbing medium with excitation through a layer of glass microspheres 200 µm in diameter [8]. The medium is a solution of ink in water with an absorption coefficient $\alpha \sim 9 \times 10^{-3}$ µm$^{-1}$ ($\lambda = 0.532$ µm). A Q-switched Nd:YAG laser with a zero transverse mode and frequency doubling. The radiation is a pulse with a duration of 15 ns. Pulse energy is 100 mJ. The beam radius at the entrance to the medium (or to the layer of spheres) is $R_{\text{beam}} \approx 750$ µm.

The scheme of medium irradiation is shown in Figure 1a. The structure of the laser beam focused by a single sphere with radius $R = 100$ µm in pure water is shown in Figure 1b. (Hereinafter, $z$ is counted from the center of the sphere.)

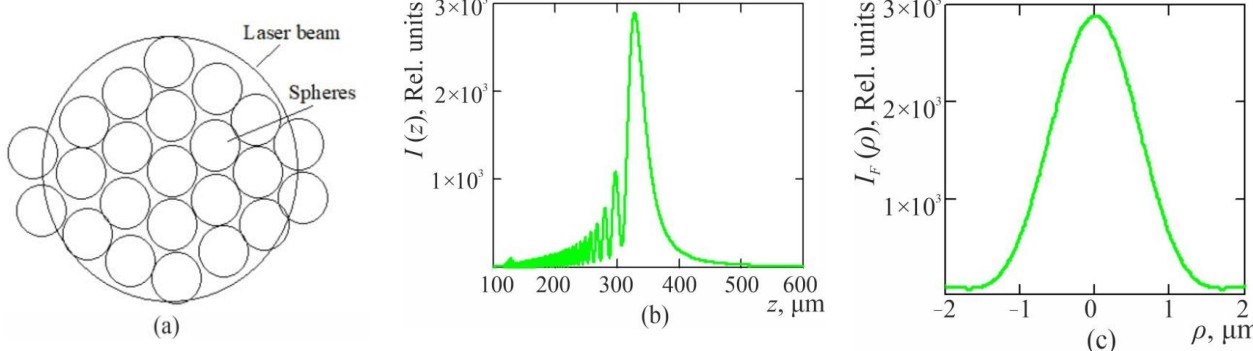

**Figure 1.** Scheme of laser beam with a 200 μm spheres coating. (**a**) Scheme of laser irradiation. The diameter of the laser beam is ≈$1.5 \times 10^3$ μm. The diameter of the spheres is ≈200 μm. (**b,c**) Structures light intensity $I(\theta, z)$ of a laser beam in pure water focused by a single sphere 200 μm in diameter: $I(0,z)$–along the beam (**b**) and $I_F(\rho)$ in geometrical focus (**c**), where $z$ is the distance along the beam and $\rho$ is the distance along the radius.

The indicatrix, the intensity distributions along the beam, the radial distribution in geometric focus, and the axial temperature distribution (neglecting the sound travel time $\tau_1 = d/c$) in a solution of ink in water are shown in Figure 2.

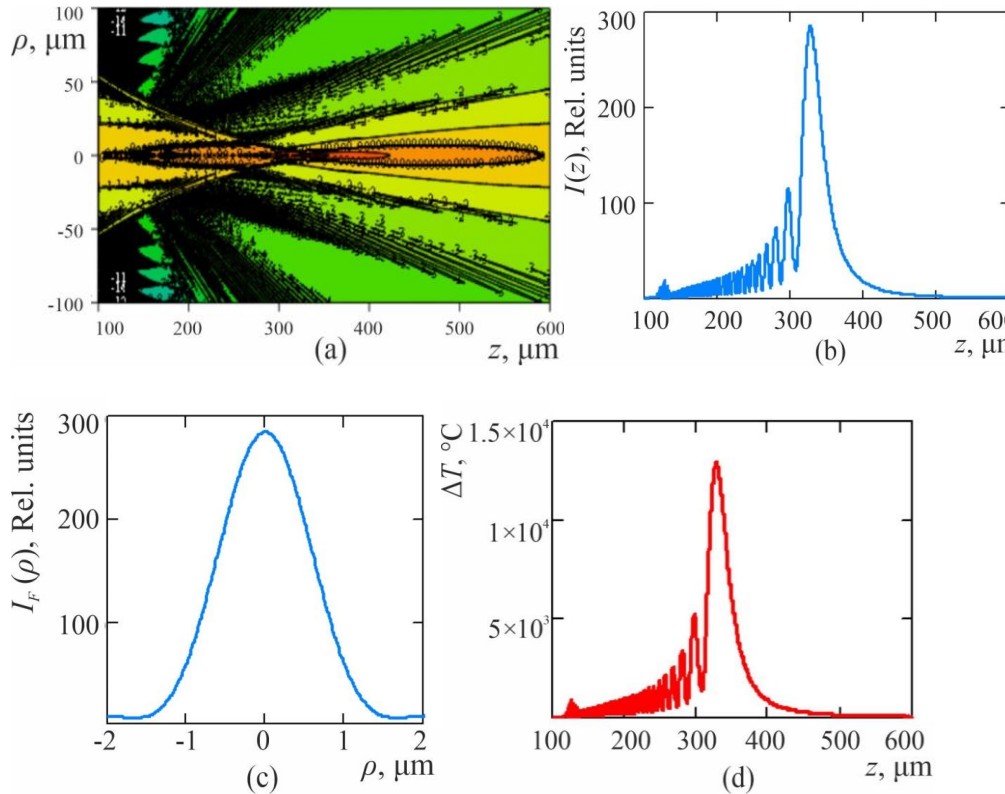

**Figure 2.** Laser beam structure [9,10] in a solution of ink in water (along the beam and in focus) focused by a single sphere 200 μm in diameter. $\alpha = 9.5 \times 10^{-3}$ μm$^{-1}$, $\lambda = 0.532$ μm, $R = 100$ μm, $\tau = 0.015$ μs. (**a**) Spatial intensity distribution, (**b**) distribution along the beam, (**c**) radial distribution in the geometric focus, (**d**) axial temperature distribution (neglecting sound travel time $\tau_1 = d/c$).

The structure of the ultrasonic field generated by a single beam with diameter $\alpha$ can be described by Formula (4).

Figure 3a shows the US spectra generated in accordance with (8) in channels with diameters of ~1000 μm (laser beam diameter), 200 μm (sphere diameter), and 0.8 μm

(diameter of the focus), calculated taking into account the spectrum of the exciting laser pulse $\tau = 0.015$ μs. Figure 3b shows the indicatrices of US at frequencies of 1 and 5 MHz, generated in a channel with a radius of 1 μm.

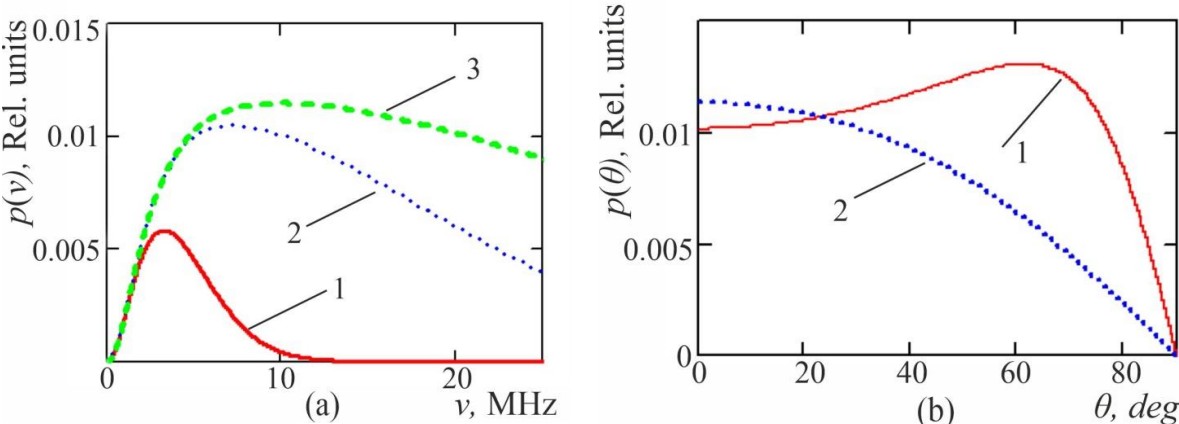

**Figure 3.** Spectra and indicatrices of US generated in channels with different radii. As was shown earlier, it follows from the data in Figure 3a that the smaller the channel diameter, the wider the generation spectrum and the wider the radiation pattern. (**a**) Generation of US pressure $p(\nu)$ spectra in channels with radii 1–500, 2–100, 3–0.4 μm. (**b**) US pressure versus angle $\theta$ at a frequency of 1 MHz (2) and 5 MHz (1) when generated in a channel with a radius of 1 μm.

Thus, as a result of focusing on a system of spheres, a two-dimensional quasi-periodic structure of heated cone-shaped objects is formed under the spheres. The diameter of these objects varies from 200 μm to a submicron value near the focus. An accurate quantitative calculation of ultrasonic generation, for example, by the FDTD method, meets certain difficulties. In addition, the idea of the excitation mechanism is lost. Therefore, here we used the simulation method. We divided the cone-shaped heated area into $i$ areas with average diameter $a_i$ and consider, in accordance with the superposition principle, that each section makes its contribution in accordance with Formula (4) and with the corresponding weight (length of the section). That is, by taking into account the additivity of various sections of the cone-shaped radiator, we make up the sum of the type:

$$S2(\vartheta, \nu) = \sum_0^i l_i \times s2(\vartheta, \nu, a_i, \tau), \tag{5}$$

where $S2(\vartheta,\nu)$ is the sound response from a single cone-shaped excitation, $s2(\theta,\nu,\rho,0.015)$ are the functions (4) describing the spectral and angular properties of the ultrasonic field according to the TE mechanism from a channel of radius $a_i$ at a pulse duration of $\tau = 0.015$ μs, and the coefficient $l_i$ is the length of the $i$-th section of the caustic corresponding to the relative length of the segment. The numerical factor in $s2$, in contrast to (4), is omitted.

Furthermore, assume for simplicity that the beam radius $\rho$ changes piecewise linearly from the sphere radius $R$ to the radius in the focal waist $a_{\min}$ in focus. Thus, in the coordinate interval $\rho = [a_{\min}; R - \delta]$, we write $S2(\vartheta, \nu, \rho, 0.015)$ as:

$$S2(\vartheta, \nu) = R^{-1} \int_{a_{min}}^{R-\delta} s2(\vartheta, \nu, a, 0.015) da + S2a(\vartheta, \nu) + S2b(\vartheta, \nu) \tag{6}$$

The first term in (6) describes the contribution from the beam region with radius from $a_{min}$ in focus to $R - \delta$, the second term corresponds to the ultrasonic response from the full laser beam with radius $a = R$ and length $\delta$, where $\delta$ is some beam length near the layer of spheres, on which the total laser beam can be considered uniform over the full diameter

of the laser beam (fiber) $R_{beam}$, and the third term corresponds to the ultrasonic response from the laser beam with radius $a_{min}$ and length ~0.01$R$.

Examples of the dependence of the function $S2(\theta,\nu)$ on the frequency $\nu$ and the observation angle $\theta$ are shown in Figure 4a,b.

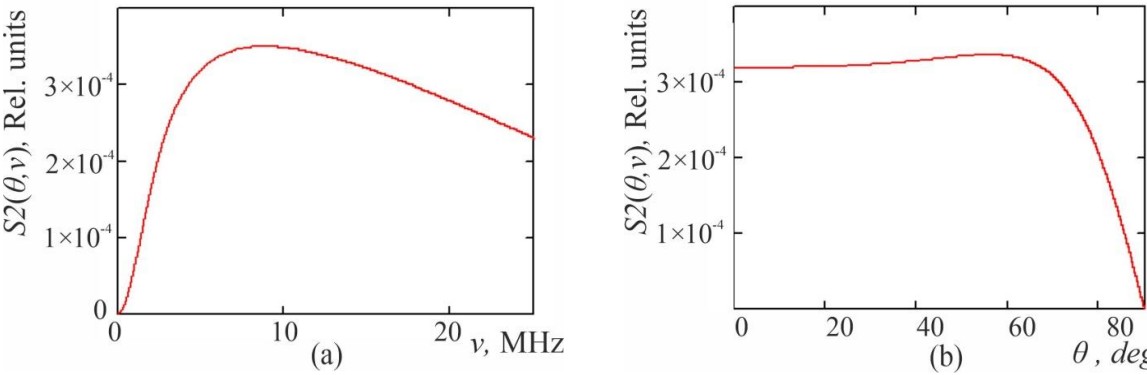

**Figure 4. (a)**. Function $S2(\theta,\nu)$ versus frequency $\nu$ at $\theta = 10°$. Sphere diameter is 200 μm. **(b)** Function $S2(\theta,\nu)$ versus angle $\theta$ at frequency $\nu = 5$ MHz. Sphere diameter is 200 μm.

The interference from different emitters was taken into account by adding the fields with the corresponding phases, taking into account that the difference in phase delays $d$ between adjacent emitters depends on the observation angle $\theta$ as $d = b \sin\theta$, where the distance between the rows $b = 180$ μm and the speed of sound in water $c = 1.5 \times 10^3$ μm/μs. The corresponding factor $P(\theta,\nu)$ according to Figure 1a, taking into account 7 rows of spheres, can be written as:

$$\begin{aligned}
P(\vartheta, v) = {}& 1\cos(-(2\pi bv/c)/\sin\vartheta) \\
& +3\cos(0(2\pi bv/c)/\sin\vartheta) + 4\cos((2\pi bv/c)/\sin\vartheta) \\
& +5\cos((4\pi bv/c)/\sin\vartheta) \\
& +4\cos((6\pi bv/c)/\sin\vartheta) + 3\cos((8\pi bv/c)/\sin\vartheta) \\
& +1\cos((10\pi bv/c)/\sin\vartheta), x
\end{aligned} \tag{7}$$

The overall result, the ultrasonic field from the beam system in the far field $S2(\theta,\nu)$ in the same units as in [8], is found as:

$$S2_1(\vartheta, v) = P(\vartheta, v)S2(\vartheta, v)L(v), \tag{8}$$

where $L(\nu)$ is laser pulse spectrum.

### 2.4.2. Small Spheres

Here we discuss the US generation in a weakly absorbing medium with excitation through a layer of polystyrene microspheres 1 μm in diameter. The medium is water with absorption $\alpha \sim 10^{-5}$ μm$^{-1}$ ($\lambda = 1.064$ μm). An Nd:YAG laser with passive Q-switching. The radiation is a train of subpulses having a duration of $\approx 0.3$ μs with a total duration of $\approx 350$ μs. The energy in a single subpulse is up to 0.005 J, in a full train up to 0.2 J, with the number of subpulses being up to 40.

The structure of the light field after the spheres is shown in Figure 5a–d [9]. This structure differs significantly from the one discussed above. The structure consists of a bulk periodic system of pellets of a heated liquid (Figure 5a).

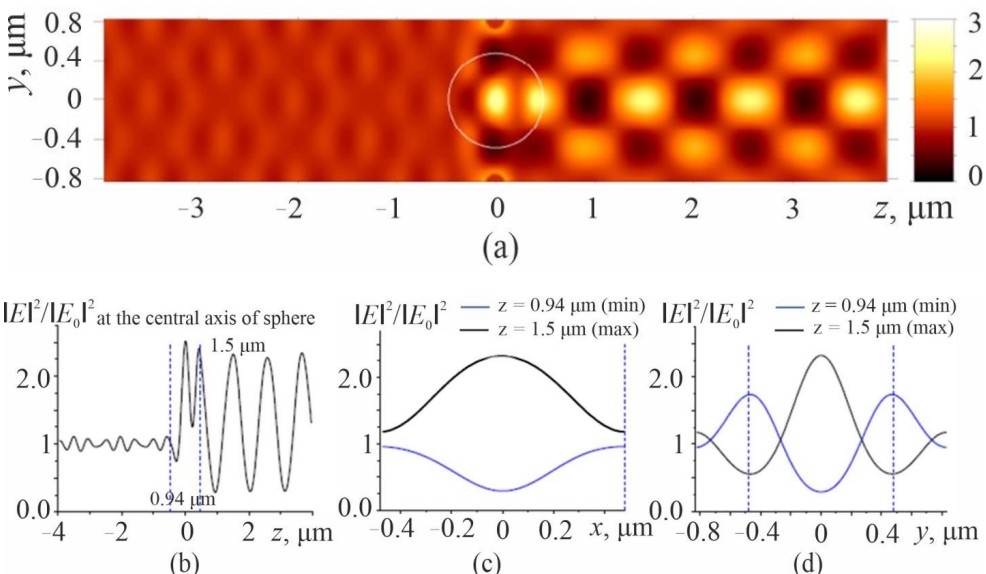

**Figure 5.** Calculated distribution of the squared electric field produced by a close-packed monolayer of polystyrene spheres (n1 = 1.58, *d* = 0.96 μm). Light ($\lambda$ = 1.064 μm) passes from the fiber ($n_{fib}$ = 1.46) to water ($n_0$ = 1.33) through the spheres. (**a**) General view of the interference pattern from 2D coating: light passes from left to right, white circle corresponds to one sphere; (**b**) field distribution along the optical *z*-axis, blue dotted lines correspond to sphere boundaries; (**c,d**) field distributions along the transverse *x* and *y* axes.

To interpret the experimental results, we consider the medium heated by the laser beam after each sphere as a "thread" of variable radius. Areas with a large radius, where the heated areas from neighboring spheres overlap, will be considered as the result of heating by a beam with a diameter of 1000 μm, i.e., with fiber diameter. The spectral sensitivity of US excitation in water in channels with radii of 500 (1), 10 (2), and 0.5 (3) μm at an angle of 80° according to (4) is shown in Figure 6.

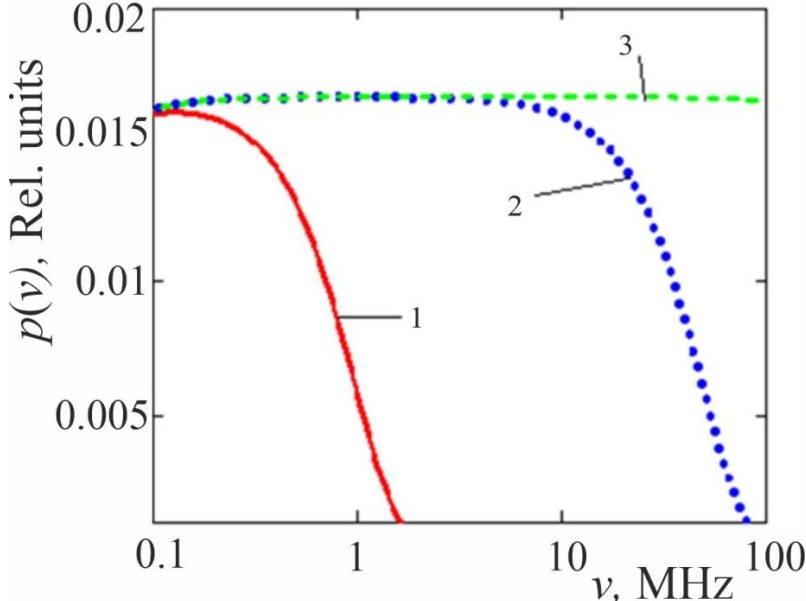

**Figure 6.** Spectral sensitivity of US excitation in water in beams with radii 500 (1), 10 (2), and 0.5 (3) μm at 80° angle.

The regions (pellets) with smaller radii will be divided, as before, into a number of regions with average diameter, and the total result of US generation $S1(\theta,\nu)$ is considered as the sum of US fields (calculated by (4)), taking into account the spectrum of the exciting laser pulse (Figure 6) in integral form of the type (6).

When calculating the resulting US from the system of pellets, we take into account that the number of spheres in the beam is up to a million pieces, and the number of active sources—pellets—is several billion. With such a number of sources, numerous violations of the periodic structure associated both with the imperfection of the technology for applying a layer of spheres and with a spread in the diameters of the spheres are inevitable [13]. Therefore, in contrast to the consideration of US from large spheres, the possibility of interference violation should be taken into account.

Let us write the interfering factor $D1(\theta,\nu)$ in the ultrasonic response in the case of complete interference from all filaments (with a distance between the rows of spheres $q \approx 0.8$ μm) similarly to (7) in the form:

$$D1(\theta,\nu) = \sum_{j=0}^{999} \left[ 2\left| \sqrt{500^2 - (500 - j))^2} \right| \right] \times \cos(2\pi q j \nu / c \times \sin\theta), \qquad (9)$$

and with random interference in the form:

$$D0(\theta,\nu) = \sqrt{ \left[ \sum_{j=0}^{999} \left[ 2\left| \sqrt{500^2 - (500 - j))^2} \right| \right] \times \cos(2\pi q j \nu / c \times \sin\theta) \right]^2 } \qquad (10)$$

The spectrum of the ultrasonic field $S_s(\vartheta,\nu)$ (in units same as in [9]) at partial interference is considered as the sum of the field with interference with the contribution $k$ and of the field without interference with the contribution $(1-k)$:

$$Ss(\vartheta,\nu) = 20 \left( \lg(S1s(\vartheta,\nu)(kD0(\vartheta,\nu) + (1-k) D1(\vartheta,\nu)) L(\nu) \right) \qquad (11)$$

Here, $L(\nu)$ is the spectrum of the exciting laser pulse.

Note that the solution of the form (11) is a solution to the US excitation equations, since it is a linear combination of exact solutions and describes the problem close to the one being solved. Therefore, we can expect that it accurately describes the main properties of the solution to the problem under consideration.

Accounting for the periodicity of pellets along the laser beam.

In this case, the total length of the exciting beam includes tens of thousands of pellets. Therefore, the interference of individual pellets is significant only at low US frequencies (kilohertz). Consequently, the frequency and angular characteristics of US in the considered range can be described mainly by the expression (11).

## 3. Results

### 3.1. Large Spheres

The final result of the simulation—the spectrum of the generated ultrasound with 200 μm spheres—is shown in Figure 7b ((3, dash-dot, red) together with the laser spectrum (1, black)) and the US spectrum (2, blue) generated from a clean fiber without spheres. Figure 7a shows the experimental spectrum from [8].

The graphs of the ultrasonic intensity versus the angle for different regions of the spectrum (1—(0–4)MHz without spheres, 2—(0–4) MHz with spheres, 3—(4–22) MHz with spheres) are plotted in Figure 8: (a) model, (b) experiment [8]. One can see a qualitative agreement between the simulation results and the experimental results–the structure of the spectrum and of angular dependences, in particular, the sharpening of the angular dependence of 4–22 MHz at irradiation through the spheres.

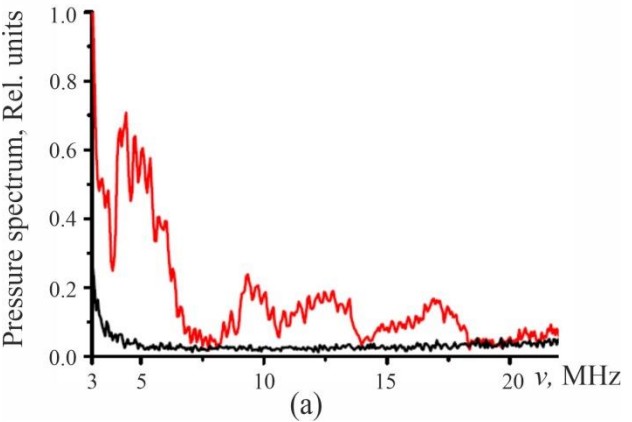
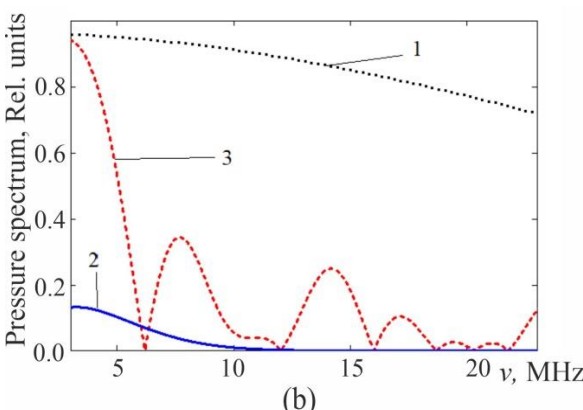

**Figure 7.** Experimental and model US spectra generated with 1μm sphere coating. (**a**) Experimental US spectra with sphere coating (red) and without coating (black) [8]. (**b**) Model US spectra: 1—spectrum of exciting laser radiation. 2—without spheres from a clean fiber spectrum; 3—spectrum with spheres; diameter of the spheres is 200 μm.

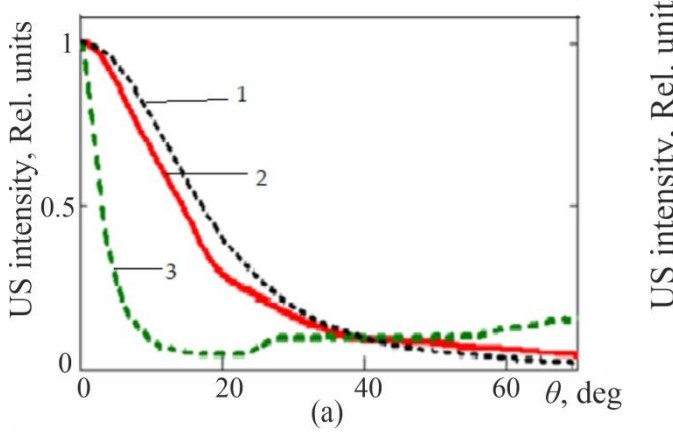
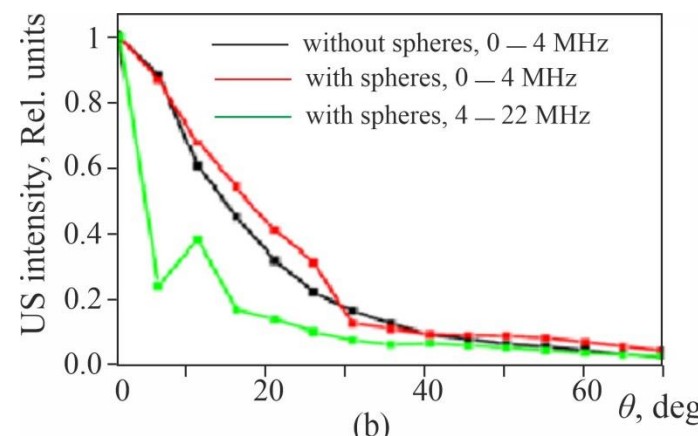

**Figure 8.** Model and experimental US intensity generated with 1 μm sphere coating versus angle $\theta$. (**a**) Model US intensity versus angle $\theta$ for various spectral regions (1–3). (**b**) Experimental US intensity versus angle $\theta$ for various spectral regions [8].

### 3.2. Small Spheres

The results of calculations for the spectrum are shown in Figure 9 together with the experimental results [9] for a coating consisting of 1 μm spheres: the green points correspond to the experimental OA spectrum from [9], red—model with chaotic interference; blue—with full interference. From the data in Figure 9, it follows that the results with chaotic interference are the closest to the experimental data, which indicates the prevalence of ultrasonic sources.

Figure 10 shows the simulated US spectrum (red), taking into account partial interference, namely, 20% of the spheres are located periodically and 80% chaotically, which is the result of calculations, experimental US spectrum (green) and the spectrum of the exciting laser pulse (blue) [9].

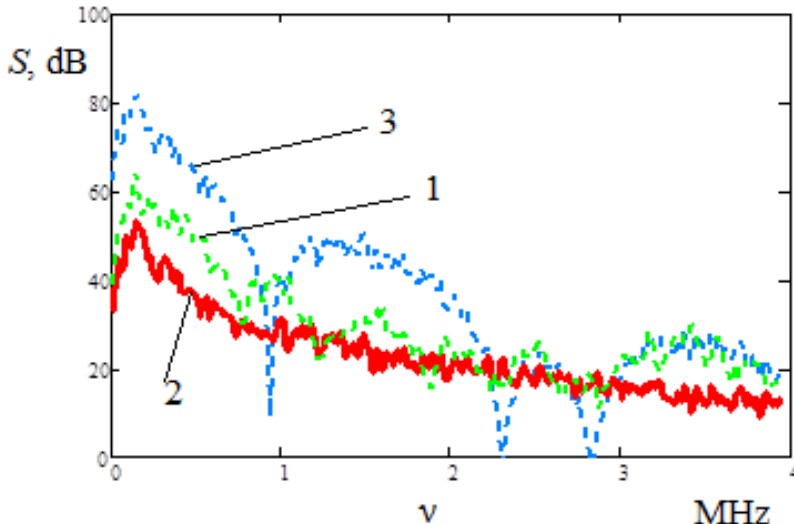

**Figure 9.** Spectra $S(\nu)$: 1—experimental OA response from [9], 2—calculation according to the model with chaotic interference; 3—calculation according to the model with full interference. The diameter of the spheres is 1 μm.

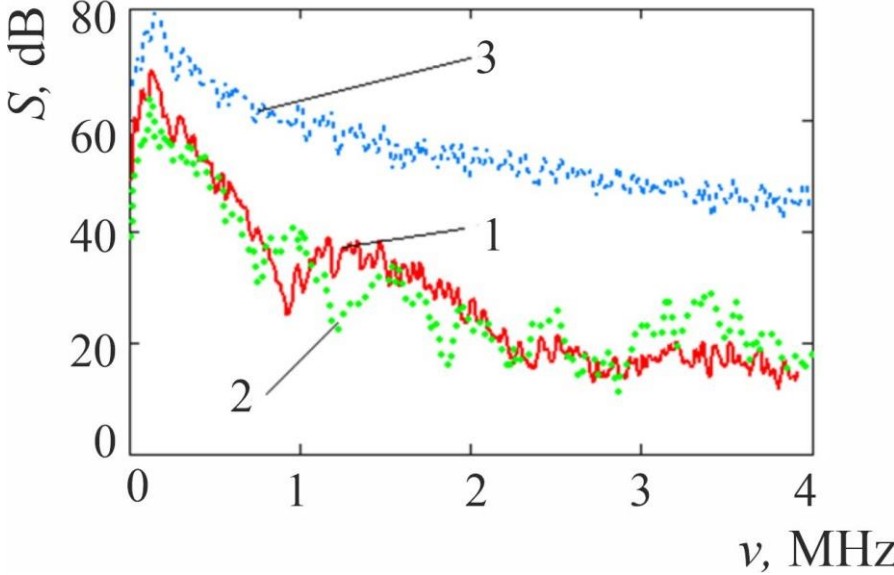

**Figure 10.** Spectra $S(\nu)$: 1—calculated US spectrum in the approximation of 20% of periodically arranged spheres, 80% arranged chaotically, 2 and 3—experimental US spectrum and the spectrum of the exciting laser pulse [9]. The diameter of the spheres is 1 μm.

The results shown in Figure 10 indicate that, despite the apparent roughness of the model under consideration, there is a fairly good agreement between the experimental and theoretical results.

## 4. Discussion

The results obtained indicate that the use of the OA transducer made of spheres really makes it possible to expand the region of acoustic generation to the region of higher frequencies.

The methodology of the model, the algorithm for studying the properties of OA converters, and the demonstrated results show the possibility of optimizing the design of the OA converter to obtain a more intense generation in the given frequency range.

The simulation performed without allowance for nonlinear effects is in a good agreement with the experiments.

Indeed, for both situations under consideration, the nature of the model US spectra (the ratio of low-frequency regions associated with the total size of the laser beam and high-frequency regions associated with the optical structure produced by the sphere coating, the presence of a characteristic interference structure in the high-frequency region of the spectrum) correlate well with the corresponding experimental data. In addition, the calculated angular dependences, on the whole, agree with the experimental ones. Some differences, for example, in the characteristic interference periods (having close magnitudes) in the spectra (Figure 7a,b and Figure 10) are explained by imperfect coating structure (dispersion of sphere diameters, irregular structure of coatings, for example, biperiodic, with vacancies). The differences in the angular dependences, in particular, the "non-smoothness" of the 4–22 MHz curve in Figure 8b may have an analogous explanation. Thus, despite its simplicity, the model gives a fairly complete picture of the processes and results of ultrasound generation with the use of radiation converters in the form of coatings from focusing spheres.

Meanwhile, the estimates show that the liquid in the focal regions can be heated to relatively high temperatures. According to the data in Figure 2d, for the experiments with 100-μm spheres and light-absorbing liquid (ink in water), the heating temperature in focus may be more than $10^4$ K (neglecting the sound travel time $\tau_1 = d/c$).

Such a water temperature can be attained only under special conditions [7,14]. In our conditions, we must take into account a short sound travel time through a heated region with diameter $r = 0.5$ μm, $\tau_1 = r/c = 3 \times 10^{-10}$ s, and pulse duration $\tau_p = 1.5 \times 10^{-8}$ s. With such a $\tau_{1/}\tau_p$ ratio, the coefficient of thermodynamic relaxation (temperature drop) can be estimated as $\zeta \approx \tau_1/\tau_p \approx 3 \times 10^{-10}/1.5 \times 10^{-8} = 2 \times 10^{-2}$.

Taking into account the temperature drop due to cooling, we obtain the distribution of the expected temperature increase in a laser jet (Figure 11). A light-absorbing liquid can be heated up to temperatures of $\approx 300$ °C in focus, but no additional effects are observed in the acoustic response. Perhaps this is due to the relatively small volume of narrow, relatively strongly superheated regions (Figure 11).

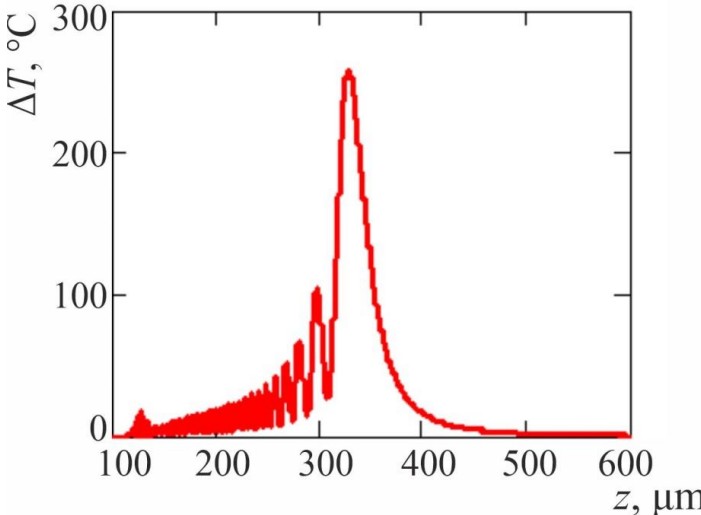

**Figure 11.** Axial distribution of the temperature increase in the beam, taking into account the temperature relaxation during the pulse. Sphere diameter is 200 μm. *z* is axial coordinate.

Meanwhile, in addition to thermal effects in the described experiments, one could also expect manifestations of other physical effects associated with phase transitions in matter, such as bubble formation, cavitation, and ablation [15,16]. Note that the latter can be both optical (the effect of optical radiation) and ultrasonic (the effect of rather powerful ultrasound on the surface of materials and microparticles in a liquid). It can be seen that in

the described experiments, at a high enough laser pump power, the spheres detach from the substrate, which we attribute to the action on the coating of the backward ultrasonic wave. These phenomena are of considerable interest for the physics of the interaction between laser radiation and high-frequency ultrasound in media and related applications and require special studies.

The authors believe that, taking into account the complex excitation geometry, the uncertainty of a number of parameters, and the simplified calculation model, a fairly satisfactory agreement between the experiment and the model was obtained for the main indicator—the spectrum of ultrasonic generation. It should be noted here that the results of the calculation are weakly dependent on the variation of the calculated parameters (for example, dividing the exciting, heated region into a number of sections) within reasonable limits.

The results of the work show the main drawback of the discussed ultrasonic generation technique, namely, a low coefficient (not higher than 10%) in the high-frequency region. This is due to the geometry of the exciting laser beam and to the small length of the section with a small diameter, in which high-frequency ultrasound is generated. In addition, the laser field in this section of the absorbing medium is significantly weakened compared to the incident field. US interference from different spheres significantly distorts the resulting spectrum. On the other hand, this makes it possible to obtain the maximum ultrasonic field in the most interesting and necessary spectral range. The conversion coefficient can be increased by transferring the narrowest section of the laser beam to the contact of the laser tool with the medium, using, for example, focusing elements, such as spheres with a high refractive index, or by making the distal end of the light guide of more complex shape. Indeed, generalizing the results of the work, we can point to the possibility of controlling the generation of ultrasound by changing the geometry of exciting optical radiation. In this regard, we can note methods for controlling a laser beam with non-imaging optics techniques [17], using optical elements of the axicon or focon type, which convert Gaussian beams into Bessel-like beams. Depending on the problem to be solved, either the laser beam as a whole may be converted to a Bessel-like beam, or its spatial system of Bessel beams may be converted using coatings from micro axicon and focon elements. Of interest are also phase coatings (such as Fresnel zones) of the distal end of the fiber.

## 5. Conclusions

The presented experimental [8,9] and theoretical (model) studies have shown that megahertz ultrasound can be excited in light-absorbing liquids by injecting laser radiation through a converter in the form of a monolayer of transparent focusing spheres. The results make it possible to evaluate the main parameters of the converters depending on the parameters of the liquid and the converter. It is demonstrated that the use of a converter allows extending the generation range to the high-frequency region.

There are still challenges related to nonlinear processes in the focal region of the spheres. It is especially interesting and important to study the processes associated with phase transitions, such as vaporization, superheated liquids, and others. The developed algorithm for calculating the spectra of OA converters and the results obtained will be useful for further study of high-temperature processes occurring when the converter focuses from spheres into a liquid. The presented results make it possible to develop such and more complex systems.

**Author Contributions:** Conceptualization, methodology, software, validation, formal analysis, investigation, resources, data curation, writing—original draft preparation, writing—review and editing, visualization, supervision, V.I.B. and V.V.K.; project administration, V.I.B.; funding acquisition, V.I.B. All authors have read and agreed to the published version of the manuscript.

**Funding:** The research was supported by the Ministry of Science and Higher Education of the Russian Federation, state assignment for the Institute of Applied Physics RAS, project 0030-2021-0012.

**Institutional Review Board Statement:** Not applicable.

**Informed Consent Statement:** Not applicable.

**Data Availability Statement:** All research results are presented in the paper.

**Acknowledgments:** The authors are grateful to N.M. Bityurin, V.A. Kamensky, A.V. Pikulin and for helpful discussions, and especially to A.V. Pikulin for computing the diffraction of light on the spheres layer.

**Conflicts of Interest:** The authors declare no conflict of interest.

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
