# Peer review of "Generation of High-Frequency Ultrasound in a Liquid upon Excitation by Laser Radiation through a Light Guide with a Converter of Transparent Spheres"

_coatings, doi:10.3390/coatings13010055_

Round 1
Reviewer 1 Report
The authors calculated the US on different parameters spheres produce by nanosecond laser. The calculated results were compared with some experimental result and a good agreement was presented. This work will help to generate the high-frequency ultrasound in liquids.
I recommend to accept it with some minor revision.
1. The resolution of the pictures such as Fig.1(b) need to be improved.
2. Maybe the experimental results and the simulation results in fig.7 can be compared in one figure. It will be more intuitive.
3. What is the reason of the different between fig.8(a) and (b) on the angular dependence of 4-22 MHz?
Author Response
The authors calculated the US on different parameters spheres produce by nanosecond laser. The calculated results were compared with some experimental result and a good agreement was presented. This work will help to generate the high-frequency ultrasound in liquids.
I recommend to accept it with some minor revision.
- The resolution of the pictures such as Fig.1(b) need to be improved.
Answer: Improved. Thanks.
- Maybe the experimental results and the simulation results in fig.7 can be compared in one figure. It will be more intuitive.
Answer: 5 curves in one figure also seem difficult to perceive. Therefore, we left 2 figures, but added a fragment (Insert 3) with comments on the differences between the experimental and model curves and possible causes. It seems to us that this should contribute to the perception of the material. Thank you.
- What is the reason of the different between fig.8(a) and (b) on the angular dependence of 4-22 MHz?
Answer: Possible reason of this different, in our opinion, is irregularity of the structure composed by spheres. We included the Insert 3 on this remark. Thank you.

Reviewer 2 Report
Your paper depicts an interesting new theory underpinned by simulations and some experimental evidence for the creation of US by laser excitation.
Although your work goes into detail about the mathematical formulation, I would like to see some more detailed assumptions about the generating laser in Chapter 2.1, where a simple Gaussian Laser beam profile is assumed. Your deprive your own work of some relevance by not extending your calculations to a more technical relevant profile, instead of the pure Gaussian. Chapter 2 also excludes the possibility of ablation which may cause the difference between your calculation and the measurements.
I) A describing paragraph that explains the laser parameters, mentions the exclusion of beam profiles, other than a Gaussian would therefore be of help.
II) Moreover if the influence of a possible ablation could be mentioned, ideally even simulated?
SO the two extra paragraphs (6-8 sentences max) could elevate the scientific relevance of the paper tremendously.
However, the general layout and errors made in the text do not allow me to accept the paper in its present form, after some major revisions, mainly in the figures you present. They are substandard for any publication and you need to address the points raised carefully. Most of the figures are too small and the annotations are not readable. Besides, the graphs themselves are depicted too thick. Y-annotation seems to be truncated in a lot of b) labels subplots and the reader is forced to understand that a) plot also supplies the Y-annotation for the b) subplot. That is bad practise. I also found a strange truncation of one author in the first citation.
So the paper cannot be accepted if the points are not addressed
In detail: On occassions it is written cm-1, e.g. line 67, 69 ,,, Please correct. I also would adivse not to typeset the equations in boldfont. The black circle bullet points are confusing in the current layout. Please correct to a more appropriate layout. Check the citation of standard SI units, e.g. nsec = ns, KHz instead of KHz
Figure 1, b too small and bad in quality, and needs improvement.
Figure 2 a) scale is not readable!
b) c) and d) need to be made clear by making them larger. E.g. scales are not
readable. Did you use a bad graphic conversion program?
Fig 3. b) No y-annotation, low quality.
Fig 4 a) and b) have different style of y-annontation though presumably showing the same, also curve to thick
Fig 5 b) c) d) All too small to read
Fig 7 b) no y-annotation, please do not simply refer to a) figure to represent the y-annotation that is not correct in a scientific publication
Fig 8 b) same as in Fig 7 b)
Fig 11) Lineplot too thick
Refrences: Check A reviewAppl. Sci. It seems to me that the author Anton Bychkov is missing? Why?
Author Response
Reviewer â„–2
Open Review
Comments and Suggestions for Authors
Your paper depicts an interesting new theory underpinned by simulations and some experimental evidence for the creation of US by laser excitation.
Although your work goes into detail about the mathematical formulation, I would like to see some more detailed assumptions about the generating laser in Chapter 2.1, where a simple Gaussian Laser beam profile is assumed. Your deprive your own work of some relevance by not extending your calculations to a more technical relevant profile, instead of the pure Gaussian. Chapter 2 also excludes the possibility of ablation which may cause the difference between your calculation and the measurements.
- I) A describing paragraph that explains the laser parameters, mentions the exclusion of beam profiles, other than a Gaussian would therefore be of help.
Answer:
Thanks for the important note. Indeed, a more important conclusion from our work is that the frequency-angular properties of the generated ultrasound can be controlled through the structure of the laser radiation. A Gaussian beam that is convenient for research may not be optimal. Perhaps for specific tasks Bessel-like, or even multimode ones will turn out to be more effective. It is also interesting to use the method of Fresnel zones. As per your comment, we have included Insert 5.
- II) Moreover if the influence of a possible ablation could be mentioned, ideally even simulated?
SO the two extra paragraphs (6-8 sentences max) could elevate the scientific relevance of the paper tremendously.
Answer: Very interesting. The ablation of materials and tissues in a liquid under the combined action of laser and ultrasonic fields is currently insufficiently studied. One of the reasons for this is the lack of effective sources of ultrasound in this area, which was the reason for this work. According to your remark, we have included in the article Insert 4. Thank you very much.
However, the general layout and errors made in the text do not allow me to accept the paper in its present form, after some major revisions, mainly in the figures you present. They are substandard for any publication and you need to address the points raised carefully. Most of the figures are too small and the annotations are not readable. Besides, the graphs themselves are depicted too thick. Y-annotation seems to be truncated in a lot of b) labels subplots and the reader is forced to understand that a) plot also supplies the Y-annotation for the b) subplot. That is bad practise. I also found a strange truncation of one author in the first citation.
Answer: We apologize for these errors. We tried to eliminate them. In particular, all drawings have been redesigned taking into account your comments on the size of drawings, captions, etc. Restored justice in relation to the author of the Ref. 1. We apologize to our colleague Anton Bychkov. It is not special. Thank you.
So the paper cannot be accepted if the points are not addressed
In detail: On occassions it is written cm-1, e.g. line 67, 69 ,,, Please correct. I also would adivse not to typeset the equations in boldfont. The black circle bullet points are confusing in the current layout. Please correct to a more appropriate layout. Check the citation of standard SI units, e.g. nsec = ns, KHz instead of KHz
Answer:Improved. Thank you.
Figure 1, b too small and bad in quality, and needs improvement.
Answer:Improved. Thank you.
Figure 2 a) scale is not readable!
- b) c) and d) need to be made clear by making them larger. E.g. scales are not
readable. Did you use a bad graphic conversion program?
Answer: Improved. Thank you.
Fig 3. b) No y-annotation, low quality.
Answer: Improved. Thank you.
Fig 4 a) and b) have different style of y-annontation though presumably showing the same, also curve to thick.
Answer: Improved. Thank you.
Fig 5 b) c) d) All too small to read.
Answer: Improved. Thank you.
Fig 7 b) no y-annotation, please do not simply refer to a) figure to represent the y-annotation that is not correct in a scientific publication.
Answer: Improved.Thank you.
Fig 8 b) same as in Fig 7 b) .
Answer: Improved. Thank you.
Fig 11) Lineplot too thick.
Improved. Thank you.
Refrences: Check A reviewAppl. Sci. It seems to me that the author Anton Bychkov is missing? Why?
Answer: It is restored justice in relation to the author of the Ref. 1 (Insert 6). We apologize to our colleague Anton Bychkov. It is not special.

Reviewer 3 Report
The material is well written. In my opinion, after some changes, it can be published.
1- The quality of the figures can be improved.
2 - The figure in 1 b is made up of two graphics, and the legend can be rewritten to make it clear what each graphic represents.
Author Response
Reviewer 3
Open Review
Comments and Suggestions for Authors
The material is well written. In my opinion, after some changes, it can be published.
1- The quality of the figures can be improved.
Unswer: Thank you. The quality of the figures was improved.
2 - The figure in 1 b is made up of two graphics, and the legend can be rewritten to make it clear what each graphic represents.
Answer: Thank you. Figures, legends were improved.

Reviewer 4 Report
Title: Generation of high-frequency ultrasound in a liquid upon excitation by laser radiation through a light guide with a converter of transparent spheres
The paper is devoted to an important subject of high-frequency MegaHertz ultrasound excitation (> 1 MHz) in biological tissues to potentially improve drug delivery and suppress the activity of microorganisms. For this goal, coatings consisting of focusing spheres on an optical fiber tip were proposed. The presented paper demonstrates a theoretical approach for calculating the ultrasound spectra depending on the sphere size, refraction index, and laser radiation parameters. In strongly and weakly absorbing media, two cases of small (of the order of wavelength) and large spheres (several hundreds of wavelength) were simulated. A comparison between theoretical and experimental results (generation up to 20 MHz) obtained in earlier published works is provided. Despite the relative roughness of the presented model, there is a fairly good agreement between the experimental and theoretical results. Prospects for developing ultrasound generation technology using fiber systems coated by transparent spheres are discussed.
However, some variables need to be introduced explicitly; there is inconsistency in numerical parameters. The authors are asked to address the following remarks:
Keywords and introduction. The "optoacousctics" term is too large and refers to any interaction between sound and light. The optoacoustics branch, which considers the laser thermo-elastic excitation of ultrasound, is officially referred to as "laser ultrasonics" or "laser ultrasound." For better visibility of the paper, it is recommended to include one of these terms in the keywords and the introduction.
Lines 108-111. "It can be seen that at a rather complex frequency, the angular structure of the generated ultrasound and its dependence on the light absorption coefficient of the medium physically indicate the result of the interference of the ultrasonic fields generated by elementary volumes of the medium heated by optical radiation."
Possibly, "at" is missing in this phrase.
Line 139. "and a is the characteristic size of the excited region". Earlier (line 103), variable a was introduced as the radius of the parallel Gaussian beam. Are these variables the same? Please, precise it explicitly.
Figure 2. Please, add labels for axes.
Line 195. Equation (8), presented on page 7, is used to calculate the results in Figure 3(a) on page 6. Perhaps the sequencing of the narrative needs to be reconsidered.
Lines 90 and 234. The same variable d is introduced twice for "characteristic size of laser beam" and "difference in phase delays."
Line 235. The variable b is not introduced. It needs to be explained why it is equal to 0.18 mm.
Figure 5 caption. "(c, d) field distributions along the transverse x-axis." figure (d) corresponds to the y-axis.
Equation (9). It needs to be explained why the exact numerical values of 500 and 999 are used. How was the distance between the rows (0.8 µm) calculated?
Figure 9 and 10 captions. "The diameter of the spheres is 0.096 μm." It was stated earlier (lines 55 and 162) that the diameter of spheres is 0.96 µm.
Discussion. Figure 11 shows a temperature increase of up to 250 K. The polystyrene melting point is 240°C. Could it be a problem for heating applications with small spheres?
Discussion. The pros and contras of large and small spheres usage should be discussed.
Everywhere in the text, diameters and radii are simultaneously used to indicate geometric parameters; the same numerical values are sometimes expressed in microns, sometimes in millimeters. Please, uniformize the numerical formats.
Author Response
Reviewer 4
Comments and Suggestions for Authors
Title: Generation of high-frequency ultrasound in a liquid upon excitation by laser radiation through a light guide with a converter of transparent spheres
The paper is devoted to an important subject of high-frequency MegaHertz ultrasound excitation (> 1 MHz) in biological tissues to potentially improve drug delivery and suppress the activity of microorganisms. For this goal, coatings consisting of focusing spheres on an optical fiber tip were proposed. The presented paper demonstrates a theoretical approach for calculating the ultrasound spectra depending on the sphere size, refraction index, and laser radiation parameters. In strongly and weakly absorbing media, two cases of small (of the order of wavelength) and large spheres (several hundreds of wavelength) were simulated. A comparison between theoretical and experimental results (generation up to 20 MHz) obtained in earlier published works is provided. Despite the relative roughness of the presented model, there is a fairly good agreement between the experimental and theoretical results. Prospects for developing ultrasound generation technology using fiber systems coated by transparent spheres are discussed.
However, some variables need to be introduced explicitly; there is inconsistency in numerical parameters. The authors are asked to address the following remarks:
Keywords and introduction. The "optoacousctics" term is too large and refers to any interaction between sound and light. The optoacoustics branch, which considers the laser thermo-elastic excitation of ultrasound, is officially referred to as "laser ultrasonics" or "laser ultrasound." For better visibility of the paper, it is recommended to include one of these terms in the keywords and the introduction.
Answer:
Thank you. "laser ultrasonics" is introduced in Keywords. Insert 1
Lines 108-111. "It can be seen that at a rather complex frequency, the angular structure of the generated ultrasound and its dependence on the light absorption coefficient of the medium physically indicate the result of the interference of the ultrasonic fields generated by elementary volumes of the medium heated by optical radiation."
Possibly, "at" is missing in this phrase.
Answer: All frequencies are real here. This phrase was wrong. We are talking that a complex frequency-angular properties of ultrasound is a result of the interference of the ultrasonic fields generated by elementary volumes of the medium heated by optical radiation. The phrase has been revised.( Improved 2)
Line 139. "and a is the characteristic size of the excited region". Earlier (line 103), variable a was introduced as the radius of the parallel Gaussian beam. Are these variables the same? Please, precise it explicitly.
Figure 2. Please, add labels for axes.
Answer: Thank you. It was done.
Line 195. Equation (8), presented on page 7, is used to calculate the results in Figure 3(a) on page 6. Perhaps the sequencing of the narrative needs to be reconsidered.
Answer: No, Fig. 3a illustrates US excited by unique parallel laser beam. Eq.8 gives US generated by system of laser beams produced by the whole spheres coating. Eq. 8 is used then for counting Figs.8-10. Thank you.
Lines 90 and 234. The same variable d is introduced twice for "characteristic size of laser beam" and "difference in phase delays."
Answer: Thank you. Improved for a in lines 88, 90.
Line 235. The variable b is not introduced. It needs to be explained why it is equal to 0.18 mm.
Answer: Insert “the distance between the rows b = 180 μm”; The value of 180 μm is close (rounded) to the row spacing in a dense flat structure for 200 μm spheres. Also taken from considerations of closeness of model and experimental results (see also Insert 3).
Figure 5 caption. "(c, d) field distributions along the transverse x-axis." figure (d) corresponds to the y-axis.
Answer: Improved for ” transverse x and y axes”.
Equation (9). It needs to be explained why the exact numerical values of 500 and 999 are used. How was the distance between the rows (0.8 µm) calculated?
Answer: Summation of rows over the area of a circle with a diameter of 1000 µm of circles with a diameter of 1 µm. Number of rows 1000, number of circles in a j row .
The value of 0.8 μm is close (rounded) to the row spacing in a dense flat structure for 0.96 μm spheres. Also taken from considerations of closeness of model and experimental results (see also Insert 3).
Figure 9 and 10 captions. "The diameter of the spheres is 0.096 μm." It was stated earlier (lines 55 and 162) that the diameter of spheres is 0.96 µm.
Answer: Thank you. The diameter of spheres is 0.96 µm. Improved.
Discussion. Figure 11 shows a temperature increase of up to 250 K. The polystyrene melting point is 240°C. Could it be a problem for heating applications with small spheres?
Answer: This temperature is reached in a small area with a diameter of fractions of microns at a istance of more than 1 micron from the spheres (see eg reference) for a short time. Therefore, Ps spheres in these experiments are not threatened. Note in this connection that glass spheres were used in the experiments with 200 μm spheres.
Discussion. The pros and contras of large and small spheres usage should be discussed.
Answer: Thank you. It makes sense to conduct such a discussion when discussing a specific technical problem: in what medium it is necessary to excite ultrasound, what to process, at what distance from the object, what frequency and indicatrix must be excited. This paper presents a tool that can help in solving this problem. Moreover, it is shown here that generalizing the results of the work, we can point to the possibility of controlling the generation of ultrasound by changing the geometry of exciting optical radiation (see Insert 5).
Everywhere in the text, diameters and radii are simultaneously used to indicate geometric parameters; the same numerical values are sometimes expressed in microns, sometimes in millimeters. Please, uniformize the numerical formats.
Answer: Thank you. Improved, geometric parameters are expressed in microns, but physical constants are expressed in m.

Round 2
Reviewer 2 Report
You certainly improved the paper concerning most of the points highlighted as reommended. Please revise again for small mistakes in English for the final submission.
Author Response
Dear Colleague,
Thank you very much for your careful reading of the manuscript. Together with a professional translator we again critically reviewed the manuscript and made the needed corrections. I will not write a list of the fixed items page by page and line by line, instead I will present them in groups.
- Figures 2,3,5 – 8 have been fixed. Decimal commas in the numbers have been replaced by dots.
- In the figures 1,3,7,8 captions, in addition to the number of the figure, captions have been added for better understanding and uniformity of style.
- As suggested by one of the reviewers, numerical parameters have been expressed in µs and µm.
- The list of references has been checked and adjusted in accordance with the Coatings standard .
- A number of poorly worded sentences have been revised aimed at their clarification and ease of understanding.
- In the Financing paragraph, the reference to one of the projects has been removed due to formal conditions.
In addition, a number of minor bugs have been found and corrected.
Thank you!
Sincerely Vladimir I. Bredikhin
